# From Molecular Insights to Clinical Perspectives in Drug-Associated Bullous Pemphigoid

**DOI:** 10.3390/ijms242316786

**Published:** 2023-11-26

**Authors:** Belen de Nicolas-Ruanes, Asuncion Ballester-Martinez, Emilio Garcia-Mouronte, Emilio Berna-Rico, Carlos Azcarraga-Llobet, Montserrat Fernandez-Guarino

**Affiliations:** Dermatology Department, Hospital Universitario Ramon y Cajal, 28034 Madrid, Spaincarlos.azcarraga95@gmail.com (C.A.-L.); montsefernandezguarino@gmail.com (M.F.-G.)

**Keywords:** autoimmune blistering diseases, bullous pemphigoid, drug-associated bullous pemphigoid, drug-induced bullous pemphigoid, dipeptidyl peptidase 4 inhibitors, gliptins, immunotherapy, immune checkpoint inhibitors, biologics, diuretics

## Abstract

Bullous pemphigoid (BP), the most common autoimmune blistering disease, is characterized by the presence of autoantibodies targeting BP180 and BP230 in the basement membrane zone. This leads to the activation of complement-dependent and independent pathways, resulting in proteolytic cleavage at the dermoepidermal junction and an eosinophilic inflammatory response. While numerous drugs have been associated with BP in the literature, causality and pathogenic mechanisms remain elusive in most cases. Dipeptidyl peptidase 4 inhibitors (DPP4i), in particular, are the most frequently reported drugs related to BP and, therefore, have been extensively investigated. They can potentially trigger BP through the impaired proteolytic degradation of BP180, combined with immune dysregulation. DPP4i-associated BP can be categorized into true drug-induced BP and drug-triggered BP, with the latter resembling classic BP. Antineoplastic immunotherapy is increasingly associated with BP, with both B and T cells involved. Other drugs, including biologics, diuretics and cardiovascular and neuropsychiatric agents, present weaker evidence and poorly understood pathogenic mechanisms. Further research is needed due to the growing incidence of BP and the increasing identification of new potential triggers.

## 1. Introduction

Bullous pemphigoid (BP) stands as the most common autoimmune blistering disease, presenting an estimated incidence ranging from 10 to 43 cases per million individuals per year [1,2]. Remarkably, this disorder exhibits a distinct predilection for the elderly population, with escalating incidence beyond the age of 70 years old [1,2,3,4,5]. According to a retrospective study conducted in the United Kingdom, the median age of BP onset was 80 years, underscoring the advanced age at which BP commonly manifests [6].

The underlying mechanisms of BP remain largely unknown. However, it seems to rely upon the interaction between predisposing and triggering factors. Predisposing elements include genetic background, age and comorbidities such as neurological conditions. Eventually, the exposure to a specific trigger, such as drugs, physical factors, vaccines, infections or transplantations, holds the potential to induce or exacerbate BP [7].

The diagnosis of BP is established through a combination of criteria, including clinical features, histopathological findings, positive direct immunofluorescence (DIF) and the detection of circulating IgG anti-basement membrane zone (BMZ) autoantibodies [8].

The classical clinical manifestations of BP consist of tense bullae appearing on erythematous urticarial skin, primarily localized on the trunk and extremity flexures, as well as in the axillary and inguinal folds. Less frequently, bullae may appear on seemingly unaffected skin, a condition referred to as “non-inflammatory BP”. Regardless of the inflammatory background, BP is characterized by its intense associated pruritus [1,8]. Mucosal involvement can be observed in up to 20% of BP patients, but is mild and predominantly affects the oral cavity [9]. Other bullous clinical variants include dyshidrosiform pemphioid, localized BP or lichen planus pemphigoides. Additionally, nonbullous presentations of BP encompass eczematous, urticarial and prurigo-like (pemphigoid nodularis) forms [5,8].

Histopathological examination usually reveals subepidermal detachment containing eosinophils, neutrophils and fibrin, alongside a dermal inflammatory infiltrate. In non-bullous forms, skin biopsy shows eosinophilic spongiosis with an eosinophilic dermal inflammatory infiltrate, although these findings might be non-specific [8]. Direct immunofluorescence (DIF) samples must be obtained from perilesional skin. The linear deposition of C3 and/or IgG along the BMZ in DIF displays the highest diagnostic sensitivity for BP (90.8%) [10].

Indirect immunofluorescence (IIF) displays a linear IgG deposition along the dermoepidermal junction, which is shown to occur on the epidermal side of the split while using salt-split human skin as a substrate. Enzyme linked immunosorbent assay (ELISA) testing can detect and quantify serum levels of anti-BP180 and anti-BP230 autoantibodies, which are usually positive in BP with ranging sensitivity [1]. A mosaic biochip designed to simultaneously detect multiple autoantibodies for the most common blistering diseases is commercially available. It has demonstrated high sensitivity and specificity, equivalent to each of its individual components, while streamlining the diagnostic process [8,11].

## 2. Pathogenesis of Bullous Pemphigoid (BP)

The pathogenesis initiates with the binding of autoantibodies against the hemidesmosomes in the basement membrane zone (BMZ) (Figure 1). This binding activates multiple pathways, both complement-mediated and non-mediated, leading to the release of cytokines and proteases and the chemotaxis of neutrophils and eosinophils. Proteolytic cleavage at the BMZ induces dermal–epidermal separation and blister formation, with the subsequent dispersion of hemidesmosome-associated protein fragments. These fragments may interact with autoreactive lymphocytes, intensifying the inflammatory response [1,5].

### 2.1. Antigenic Targets

In bullous pemphigoid, we usually find autoantibodies against two principal hemidesmosomal proteins: bullous pemphigoid antigen 2 (BPAg2) and bullous pemphigoid antigen 1 (BPAg1). BPAg2 is a 180 kilodalton transmembrane protein, also known as bullous pemphigoid 180 (BP-180) or collagen XVII. BPAg1 is a 230 kilodalton intracellular hemidesmosomal protein, so it is also referred to as bullous pemphigoid antigen 230 (BP230) [12].

#### 2.1.1. BP180

BP180 is a morphologically complex transmembrane protein (Figure 2). It is composed of a globular intracellular domain in the amino-terminal and a large extracellular segment (or ectodomain) in the carboxyl-terminal that encompass the lamina lucida in the dermoepidermal junction and expands into the lamina densa [13]. The ectodomain is composed of 15 collagenous domains interspersed with 16 non-collagenous (NC) domains, each designated in sequential order starting from the carboxyl-terminal (NC1, NC2 … NC16) [14].

The NC16A domain consists of an extracellular juxtamembranous region and contains the major pathogenic epitope for BP. Thus, most common commercialized ELISAs for BP diagnosis use a recombinant NC16A protein to detect and quantify BP autoantibodies [15] and 84–90% of BP sera react with the NC16A domain [16,17]. It is important to note that anti-BP180 antibodies can also recognize other epitopes on BP180 beyond the NC16A domain, extending into the midportion or carboxyl-terminal regions of the ectodomain [15]. Izumi et al. reported that these cases with non-NC16A anti-BP180 antibodies displayed a non-inflammatory phenotype with less erythema and a diminished eosinophilic infiltrate and were more likely to respond to corticosteroid treatment [18].

One of the hypotheses that has been put forth to explain the disease revolves around impaired proteolytic degradation. The physiological shedding of BP180 by serine proteases, including plasmin, exposes new antigens and generates neoepitopes, which could serve as targets for blister-inducing antibodies. The principal site of degradation for the BP180 molecule is the juxtamembranous domain NC16A, aligning with the major pathogenic epitope of BP [18,19]. Other proteinases, such as A Disintegrin and Metalloproteases (ADAMs) and Granzyme B (GzmB), might also contribute to the generation of neoepitopes and the onset of BP through BP180 cleavage. In fact, GzmB expression is upregulated with age, which could help to explain its role in this age-related autoimmune blistering disorder [20].

BP180 has also been demonstrated to be present in extracutaneous tissues, such as in various neuroanatomical regions in the brain [21], and as a component of the glomerular filtration barrier in the kidneys [22]. However, its precise function and potential role in neurodegenerative disorders or renal diseases remains to be elucidated [21,22,23].

#### 2.1.2. BP230

BP230 is an intracellular component of hemidesmosomes and is part of the plakine family. Anti-BP230 antibodies mainly bind to the globular carboxyl-terminal domain and are detected in approximately 60–70% of BP serum samples [24]. Given the fully intracellular location of BP230, the accessibility of autoantibodies to this antigen is potentially limited. Consequently, it remains uncertain whether they exert a pathogenic role in BP or merely appear as a secondary event linked to keratinocyte injury (occurring as byproducts of epitope spreading associated with disease extension) [16].

Nonetheless, anti-BP230 antibodies have been associated with the appearance of non-bullous pemphigoid [25], whereas its absence may correlate with mucosal involvement in BP patients [9]. As a result, anti-BP230 antibodies might contribute to some extent in the development of BP; however, the precise mechanism and significance remain unclear.

### 2.2. Hypothesis of Blister Formation

It is well accepted that bullous pemphigoid arises from a loss of immune tolerance, resulting in the production of autoantibodies against BP180 and BP230. These antibodies trigger an inflammatory reaction, attracting numerous neutrophils, eosinophils and mast cells, which migrate to the dermis and release a wide range of cytokines and proteases, responsible for dermoepidermal cleavage and blister formation [26].

Until the last decade, a complement was believed to be a prerequisite for blister formation by autoantibodies. Complement components are present along the dermoepidermal junction in patients with BP, as demonstrated with direct immunofluorescence (DIF), which shows linear C3 deposition in 83–84% of BP cases [27,28]. Complement proteins are present also in the blister fluid of BP patients [29]. Furthermore, complement activation by autoantibodies may correlate with disease activity, as demonstrated in laboratory and clinical studies [28,30]. All this evidence suggests a potential pathogenic role of the complement system in BP development.

Nonetheless, more recent studies have questioned its major pathogenic role in bullous pemphigoid, proposing the existence of complement-independent mechanisms mediating blister formation [28,30]. In animal models, Ujiie et al. demonstrated that the passive transfer of BP autoantibodies induced blister formation in C3-deficient humanized mice, despite not being able to activate the complement cascade [31]. Furthermore, it should be noted that if DIF shows a C3 deposition in 83–84% of patients, then 16–17% of BP patients do not present this complement protein along the dermoepidermal junction and thus they might not be mediated by the complement system [27,28]. In these cases, IgG4 antibodies are the dominant IgG subclass, which are not able to activate the complement cascade [32].

#### 2.2.1. Complement-Dependent Immune Response

The IgG1 and IgG3 antibodies bind to BP180, consequently initiating the activation of the complement cascade (Figure 3). The resulting anaphylatoxins C3a and C5a induce the chemotaxis and degranulation of neutrophils, eosinophils and mast cells. Neutrophils release proteolytic enzymes, including neutrophil elastase (NE) and matrix metalloproteinase 9 (MMP9), leading to the degradation of BP180 and subsequently weakening basal cells’ adhesion to the basement membrane zone (BMZ). Simultaneously, mast cells secrete IL-8, which amplifies the neutrophilic infiltration, and numerous proinflammatory cytokines that recruit additional eosinophils. Upon reaching the BMZ, migrated eosinophils discharge their granule proteins, culminating in subepidermal blistering [12,33]. In addition to NE and MMP9, other proteases may potentially contribute to dermal–epidermal cleavage, as evidenced by studies detecting plasmin in BP blister fluid [29,34]. The binding of IgG to BP180 on keratinocytes could induce the liberation of tissue-type plasminogen activator (tPA), thereby catalyzing the conversion of plasminogen into active plasmin [18,34].

#### 2.2.2. Complement-Independent Immune Response

The binding of autoantibodies to BP180 in hemidesmosomes results in the internalization of BP180 into basal keratinocytes (Figure 4) [35], so the adhesive strength of the dermoepidermal junction decreases. This appears to be an early event in disease pathogenesis, followed by an inflammatory response that finally causes dermoepidermal separation [26,36]. BP180 internalization occurs through the micropinocytosis pathway [36]. Afterwards, it remains unclear whether they are degraded in lysosomes (as macropinosomes usually do) or if it is mediated by ubiquitylation and proteasomal degradation [14].

Following the interaction between anti-BP180 antibodies and BP180 ectodomain, keratinocytes release proinflammatory cytokines such as IL-6 and IL-8 [37], possibly mediated by the upregulation of NF-kappa beta and STAT3 [38]. These cytokines and chemokines attract eosinophils and neutrophils, responsible for the inflammatory reaction [39].

IgG4 has a very limited ability to fix complements and has been reported by some studies as the predominant subclass of autoantibodies in BP, followed by IgG1 and IgG3 [40,41]. The dominance of IgG4 is more evident in the early stages of BP, suggesting that IgG4 may play a pathogenic role primarily in the initiation of the immune response [41].

The balance between the contributions of complement-dependent IgG1 and complement-independent IgG4 might explain the clinical diversity that we can find in BP. Certain authors support that C3-positive pemphigoids in DIF are mediated by IgG1/IgG3 and IgG4, and they would present as the classic BP with urticarial rash and worse clinical severity. On the other hand, C3-negative pemphigoids are IgG4-dominant and tend to have a non-inflammatory phenotype with milder severity [39].

Another immunoglobulin with little ability to activate complements is IgE, which is increasingly being linked to the pathogenesis of BP. Anti-BP180 IgE autoantibodies are detected in the majority of BP sera and are correlated with disease activity [42,43]. BP180–IgE complexes adhere to the keratinocyte basement membrane and bind with the FcεR1 receptors present on eosinophils, mast cells and basophils. This interaction triggers the release of proteases such as MMP9, eosinophil granule proteins and eosinophil extracellular traps. MMP9 degrades BP180, thereby contributing to dermal–epidermal separation. Eosinophils also secrete interleukin 31, which is directly related to pruritus in BP. In response to eosinophil granule proteins, keratinocytes release cytokines such as IL-5, eotaxin-1 and chemokine ligand 5 (CCL5). This cyclical process amplifies tissue eosinophilia and promotes eosinophilic spongiosis [26,27,44]. These facts support the relevance of complement-independent Th2-mediated pathways in the pathogenesis of BP.

Hence, it is plausible that both complement-dependent and independent mechanisms play a collaborative role in triggering and perpetuating bullous pemphigoid [45].

### 2.3. Breakdown of Self-Tolerance

The fundamental initial process in the development of bullous pemphigoid is the generation of autoantibodies targeting hemidesmosomal proteins. FoxP3+ regulatory T cells (FoxP3+ Treg) represent the pivotal cell population for self-tolerance maintenance, since they are responsible for suppressing excessive autoantibody production [16]. However, the scientific literature exhibits contradictory results in this regard. Some authors have documented decreased FoxP3+ Treg cells among BP patients [46,47], whereas other authors have identified a substantial increase [48]. These differences may stem from a selection bias associated with Treg markers. Specifically, Muramatsu et al. reported that total Treg cells are increased in classic BP patients before treatment, possibly secondary to the inflammatory background, but significantly decrease after corticosteroid treatment. This finding can be attributed to the inhibition of IL-2 by corticosteroids, which is required for the maintenance of Treg cells. Alternatively, corticosteroid treatment might suppress autoreactive T cells and therefore effector Treg cells would consequently decrease as they are no longer needed [49].

Nevertheless, the dysfunction of Treg cells has been identified in BP [50]. This malfunction can result in the suppression of self-tolerance and subsequently the formation of autoreactive T helper 2 (Th2) lymphocytes mediated by STAT6. Autoreactive Th2 cells are able to activate and sensitize B cells and generate antibodies against self-components [16].

### 2.4. Epitope Spreading

Epitope spreading (ES) is a phenomenon in which the immune responses of T and/or B cells extend from the original dominant epitope to other secondary epitopes as time progresses. These new epitopes may be located on the same autoantigen (intramolecular epitope spreading) or on different antigens within the same anatomical site [51].

It is widely recognized that ES is a frequent event in the development of BP. In vivo studies using murine models have demonstrated that IgG antibodies targeting BP180 initially react to epitopes situated within the ectodomain and, subsequently, they react to other extracellular and intracellular domains over time (Figure 5) [52,53]. However, this immunological reaction is not solely confined to antigens localized within BP180; rather, it progressively spreads over time to other molecules, including BP230 [53]. Furthermore, in a prospective multicenter study, ES was observed in 49% of patients following a 1-year observational period. These events exhibited a distinct propensity to occur during the early stages of the disease [54]. All these findings suggest that NC16A recognition in the BP180 ectodomain is an early event, succeeded by intra- and intermolecular ES events. These sequential occurrences collaboratively mold the individual course of each patient with BP [51].

The concept of epitope spreading has been suggested as an explanation for those cases in which BP develops in the setting of other diseases [53]. For example, the basement membrane zone disruption in oral lichen planus might expose hemidesmosome proteins and then trigger the autoimmune humoral response responsible for lichen planus pemphigoides [55]. BP may also develop after radiation therapy, possibly through the exposure of BMZ antigens during the course of the treatment [56]. Finally, ES from brain BP180 due to neurologic damage has also been proposed to partially explain the relationship between BP and certain neurocognitive diseases [57]. Although BP180 is diffusely expressed within the central nervous system, a recent study has revealed that it is not expressed in the hippocampus, which is the main area affected in neurocognitive disorders [23]. This underscores the need for future research to elucidate the intricate connection between neurological disorders and BP.

## 3. General Aspects of Drug-Associated Bullous Pemphigoid (DABP)

### 3.1. Drugs Related to DABP

The first case of drug-associated bullous pemphigoid (DABP) was reported in an 11-year-old patient receiving treatment with salicylazosulphapyridine [58]. Subsequently, a wide range of drugs have been linked to the pathogenesis of this disease.

According to the Naranjo Adverse Reaction Probability Scale [59] and the Karch–Lasagna algorithm [60], most bullous pemphigoid cases could be categorized as having a “probable” association with a drug regarding the temporal relationship, the available literature and the absence of alternative causes. However, while these scales are useful in assessing general drug reactions, their significance appears diminished when applied to the identification of potential triggers in drug-associated bullous pemphigoid. In contrast, Tan et al. proposed specific criteria to consider a drug as a potential trigger for BP. These criteria include the drug’s initiation within the preceding year, a treatment duration of more than 2 weeks and drug continuation until at least 1 month before the diagnosis of BP [61].

Medications most frequently linked to BP include dipeptidyl peptidase 4 inhibitors (DPP4i), diuretics, neuroleptics, antibiotics, monoclonal antibodies against anti-tumor necrosis factor (TNF)-α, immune checkpoint inhibitors targeting programmed cell death protein 1 (PD-1) and its ligand (PD-L1), non-steroidal anti-inflammatory drugs (NSAID) and antihypertensive drugs. However, the list is exponentially growing (Table 1). BP has even been reported to develop after the application of certain topical drugs, inducing some form of “contact pemphigoid” [62]. However, the potential of topical agents to trigger BP remains controversial, as direct associations are not well established in most cases [63].

Verheyden et al. conducted a systematic review of drug-associated bullous pemphigoid and consequently developed a diagrammatic summary of the strength of supporting evidence for each drug. Within this framework, the evidence was stronger for DPP4i, followed by immune checkpoint inhibitors PD-1/PD-L1, loop diuretics, penicillins, NSAIDs, thiazides and psoralens with ultraviolet A phototherapy [4]. Additionally, Liu et al. recently published a meta-analysis of case–control studies, in which they found a significant association between BP and the prior use of DPP4i (odds ratio [OR] 1.92), aldosterone antagonists (OR 1.75), anticholinergics (OR 3.12) and dopaminergic medications (OR 2.03) [64].

Nonetheless, the majority of these associations are predominantly drawn from case reports, relying on factors such as temporal correlation or similarity to previously reported cases. As a result, the levels of evidence for most of the suspected medications are low due to the absence of controlled studies [64]. Furthermore, these clinical associations are subject to various confounding elements, including the prevalent polypharmacy among elderly individuals and the common use of over-the-counter drugs that are seldom reported to healthcare professionals. Unfortunately, ethical and safety concerns make it infeasible to rechallenge patients in order to definitively confirm the presumed link between BP and drug exposure [4,63].

As ongoing research continues to unravel the pathogenesis and natural history of DABP, clinicians should be aware of this association in order to identify and treat potential cases of DABP early on [4,63]. Drug discontinuation in DABP might lead to a reduction in the need for immunosuppression and a better prognosis when compared to missed DABP [65].

### 3.2. Pathogenic Mechanisms

Increasing interest is being directed towards the research of DABP pathogenesis, yet a precise understanding of the underlying mechanisms is still lacking. Drugs are thought to act as triggering factors in patients with an underlying genetic susceptibility. Various studies have suggested a potential correlation between DABP and specific major histocompatibility complex (MHC) class II alleles, since they could facilitate the presentation of BMZ autoantigens to T cells [4,7].

It has been hypothesized that the pathogenesis of DABP might be explained by the interaction of several mechanisms (Figure 6).

“Two-step” theory: the interplay between two drugs with analogous molecular structures and their interaction with the immune system might represent the first and second “hits” required to initiate and amplify the immune response [4,7,63].Molecular mimicry: many drugs bind to RNA and proteins in a way that closely resembles the interaction pattern observed with viruses. This similarity raises the possibility that these drugs might be erroneously recognized as microbial antigens. The immune system’s misidentification of drugs in predisposed individuals could result in the activation of CD4+ T cells and the subsequent initiation of the autoimmune cascade [7,63].Antigenic haptens: some drugs may have the ability to function as antigenic haptens that can bind to and modify protein molecules within the lamina lucida of the BMZ. Such interactions might induce the modification of their antigenic properties, thereby acting as neoantigens. Alternatively, this phenomenon could lead to the exposure of a previously hidden antigenic site, supporting the drug-triggering epitope spreading theory [4,7,63,66].Direct immune dysregulation: drugs may cause immune reorganization, disrupting the endogenous regulatory processes that prevent the development of several diseases. Alterations in T-regulatory cell functions may suppress “forbidden” B cell clones and then result in the release of autoantibodies against the BMZ [63,66].Non-immunological mechanisms: thiol-containing drugs may directly interact with the sulfhydryl groups present in the BMZ proteins and subsequently disrupt the dermoepidermal junction without the involvement of immunological mechanisms. However, this dermoepidermal cleavage may also expose new, hidden antigenic sites [7,63].

Furthermore, Verheyden et al. [4] recently proposed that drugs related to BP may be categorized according to their chemical structure as thiol-based, phenol-based and non-thiol non-phenol-based drugs.

Thiol-based drugs: they might induce BP acting as haptens or directly disrupting the dermoepidermal junction, as previously described. Moreover, penicillamine, a specific thiol-based drug, could decrease the activity of T-regulatory cells [62]. Many drugs, such as furosemide, hydrochlorothiazide, spironolactone, penicillins or sulfasalazine, contain sulfur atoms within their molecules, yet not as part of a thiol group. However, it is hypothesized that they may be able to form thiol groups during their metabolism, thereby inducing BP through a similar mechanism to thiol-based drugs [67].Phenol-based drugs: these medications incorporate a phenyl group in their molecular structure and are thought to interfere with the integrity of the BMZ, consequently revealing hidden epitopes. Examples of these phenol drugs are non-steroidal anti-inflammatory drugs (NSAID), cephalosporins, angiotensin II receptor blockers (ARB) and selective serotonin reuptake inhibitors (SSRI).Non-thiol non-phenol-based drugs: the number of these drugs is continuously growing, although the precise underlying mechanisms remain largely undefined.

### 3.3. General Differences between DABP and Idiopathic BP

#### 3.3.1. Clinical Differences

Patients diagnosed with DABP are often younger than those affected by the idiopathic form [4,63]. In contrast to classic BP, the clinical manifestations of DABP may be more heterogenous, resembling other conditions like erythema multiforme or pemphigus, which often delays the diagnosis [4,68]. Lesions typically manifest as tense bullae on seemingly normal skin or, more infrequently, on an erythematous or urticarial base [63].

The natural course of DABP remains somewhat uncertain, although there have been recognized two variants based on their clinical history. The first is an acute, self-limited form characterized by definitive resolution upon discontinuation of the suspected drug. This form can be genuinely categorized as a drug reaction (true drug-induced bullous pemphigoid). Conversely, the second form presents a chronic and severe course, similar to classic bullous pemphigoid. It can persist even after the suspected drug is withdrawn and may require a prolonged treatment (drug-triggered bullous pemphigoid) [66].

#### 3.3.2. Histological and Laboratory Differences

Despite all the extensive research conducted in bullous pemphigoid, no specific antigens for DABP have been identified. Therefore, it is believed that the autoantigens involved might align with those identified in idiopathic BP. Direct and indirect immunofluorescence typically exhibit similar patterns to those of idiopathic BP [4,63].

The typical histological findings in DABP encompass the presence of intraepidermal vesicles, necrotic keratinocytes and a prominent eosinophilic infiltrate, with occasional thrombus formation. However, idiopathic BP typically lacks intraepidermal vesicles, necrotic keratinocytes and thrombi, and the eosinophilic infiltrate is usually milder. Marked eosinophilia in serum is frequently observed in DABP cases [4,63].

## 4. Dipeptidyl Peptidase 4 Inhibitor-Associated Bullous Pemphigoid (DPP4i-BP)

Dipeptidyl peptidase 4 inhibitors (DPP4i), also known as gliptins, constitute a class of incretin-based drugs indicated for the treatment of type 2 diabetes mellitus. DPP4i suppress the enzyme dipeptidyl peptidase 4 (DPP4), which is responsible for the degradation of the incretin hormone glucagon-like peptide-1 (GLP-1). This inhibition results in the secretion of insulin and the reduction of glucagon [69]. Their favorable safety profile, even in patients with progressive renal insufficiency, the low risk of hypoglycemia and their oral administration have led to the frequent use of these agents among elderly patients [70]. Despite their good tolerability profile, in the last few years, they have been increasingly associated with the development of bullous pemphigoid.

### 4.1. Epidemiology of DPP4i-BP

#### 4.1.1. General Risk of BP Development

The correlation between gliptin treatment and the development of BP initially emerged from anecdotal case reports. Subsequently, these associations underwent thorough examination through the analysis of two national pharmacovigilance databases [71,72], in addition to numerous observational controlled studies [73,74,75,76,77,78].

In a systematic review and meta-analysis published in 2018, the odds ratio for BP among patients receiving any DPP4i ranged from 1.27 to 3.45, with a pooled OR of 3.16 (95% CI 2.57–3.89; I2 = 36.09%; *p* = 0.196), so it can be concluded that gliptin intake might be associated with a threefold increased risk of developing BP [76]. However, this OR might have been overestimated, as some subsequent, large, nationwide case–control studies have reported milder associations following multivariant analysis (adjusted OR 1.58 [78]–1.86 [79], both statistically significant).

Notably, Kawaguchi et al. retrospectively analyzed the number of patients who had been first prescribed a DPP4i and all BP newly diagnosed cases at their medical facility for a study period of four years. They reported an incidence rate of BP among all patients taking DPP4i of 0.0859%. When stratified by specific DPP4i, BP was more incident in patients taking vildagliptin (0.292%), followed by linagliptin (0.076%), sitagliptin (0.059%) and alogliptin (0.052%) [75].

#### 4.1.2. Risk of BP Development in Patients with Diabetes Mellitus in Absence of DPP4i

Even in the absence of DPP4i use, diabetes mellitus is found to be associated with bullous pemphigoid, exhibiting twofold higher incidence compared to the general population. The linkage between BP and diabetes might be attributed to the generation of autoantibodies due to the augmented skin fragility in diabetic patients, or as a consequence of the nonenzymatic glycosylation of dermal proteins [80].

Finnish nationwide studies have shown that the use of no other antidiabetic drugs, including metformin [73], thiazolidinediones, sulfonylureas and repaglinide [81], was associated with an increased risk of BP. This implies that, in BP cases diagnosed during metformin–DPP4i combination therapy, metformin treatment can be safely continued, while careful consideration should be given to the withdrawal of the gliptin component [73]. However, as these studies were conducted in 2018, new-generation diabetic medications were being taken by very few patients. In more recent studies, a possible association has been reported with previous exposure to a sodium glucose cotransporter 2 inhibitor and glucagon-like peptide 1 receptor agonist, suggesting the need for further large-scale epidemiological studies in the future [82,83].

#### 4.1.3. Risk of BP Development in Patients with Diabetes Mellitus in Absence of DPP4i

Vildagliptin stands as the main DPP4i agent implicated in the development of BP across several studies that have included a subgroup analysis [71,72,76,78,79,84]. In a meta-analysis conducted by Kridin et al., the use of vildagliptin was reported to be associated with a ten-fold increased risk of BP, with a pooled OR of 10.16 (95% CI, 6.74–15.33; I2 = 0%; *p* = 0.702) [76]. In a more recent population-based, nested case–control study, also conducted by Kridin et al., vildagliptin again exhibited the strongest association, albeit with reduced odds (OR 3.40; 95% CI, 2.69–4.29; *p* < 0.001) [79]. Notably, vildagliptin displays relatively lower selectivity for the DPP4 enzyme when compared to other family members like DPP-8 and DPP-9 [85], which are recognized to retain procaspase-1. Consequently, it has been hypothesized that the off-target inhibition of DPP-8 and DPP-9 might trigger the activation of the inflammasome–caspase-1 pathway, potentially contributing to the pathogenesis of BP [78].

The sub-analysis of other individual DPP4i also revealed a significant association with BP development, suggesting the presence of a class effect [72]. Nevertheless, there exists greater variability in the results across them, ranging from some studies reporting a six-fold rise in the probability of BP with linagliptin [76] to others demonstrating a significant association with sitagliptin exposure [78,79].

The larger sample size offered by a retrospective population-based study conducted in Japan enabled a more comprehensive evaluation of the risk linked to each individual DPP4i. According to their findings, all daily DPP4i (alogliptin, anagliptin, linagliptin, saxagliptin, sitagliptin, teneligliptin and vildagliptin), as well as the available weekly DPP4i (omarigliptin and trelagliptin), exhibited a significant association with the risk of developing BP. Interestingly, both categories displayed roughly similar levels of risk [77].

#### 4.1.4. Latency Period

Most studies agree on the long latency period that typically elapses between DPP4i initiation and the onset of BP. In the majority of these studies, the median latency time was approximately 6 to 8 months, ranging from as short as 10 days to more than 3 years [71,72,73,74,75]. However, a recent duration–response analysis conducted by Kridin et al. revealed that the highest probability of BP onset occurred 1–2 years following DPP4i initiation, with a median latency of 3.3 years and a continuous, statistically significant risk for BP development extending beyond 6 years from the beginning of the treatment [79]. The delayed onset of BP suggests that additional factors besides the drug itself are required to break the tolerance to BP180 and to trigger the autoimmune response.

Nevertheless, Kuwata et al. recently reported that the risk associated with DPP-4i use was restricted to a span of 3 months following the first use. The risk of developing BP was most pronounced within the initial 30 days after the first administration, with a gradual decline until no risk was observed after 90 days. They hypothesized that BP cases that develop within 3 months of initiating DPP4i might involve different pathogenic mechanisms compared to cases that emerge later [77].

### 4.2. Pathogenesis of DPP4i-BP

#### 4.2.1. Genetic Predisposition

Ujiie et al. reported that 86% of their non-inflammatory DPP4i-BP Japanese patient cohort exhibited the HLA-DQB1*03:01 haplotype, compared to DPP4i-treated healthy controls, where this allele was found in only 31% of them [86]. Interestingly, this allele is also known to be associated with mucous membrane pemphigoid in Caucasian patients [87]. In a study conducted by Lindgren et al. in Finland, it was concluded that the HLA-DQB1*03:01 allele was more commonly present among BP patients when compared to the control population. However, this study failed to find significant differences in HLA haplotypes between DPP4i-associated and non-DPP4i-associated bullous pemphigoid cases [88].

#### 4.2.2. DPP4 Functions

DPP4, also known as CD26, is widely expressed in various tissues, including keratinocytes and immune T CD4+ cells (Figure 7). It is therefore reasonable to hypothesize that the inhibition of CD26 on T cells could potentially dysregulate the immune system. CD26 expression is known to be increased in various immunomediated diseases, such as psoriasis and atopic dermatitis [89] Recently, it has been reported that the cutaneous expression of CD26 is also upregulated in BP patients, regardless of their previous gliptin exposure [88].

The inhibition of DPP4 in keratinocytes may release eotaxin (CCL11) and other proinflammatory cytokines that promote dermal eosinophil recruitment, which is a common histopathologic feature of BP cases [90]. Conversely, Japanese studies have demonstrated that periblister eosinophil infiltration appears to be significantly lower in non-inflammatory DPP4i-BP than in inflammatory DPP4i-BP [18,86].

Furthermore, DPP4 acts as a surface cell plasminogen receptor that converts plasminogen into plasmin (Figure 7) [91]. Plasmin is a serine protease that can be detected in BP blister fluid that, in conjunction with other proteases, cleaves the BP180 molecule within the juxtamembranous NC16A domain. This initial N-terminal cleavage results in the production of the LAD-1 fragment (120 kDa), potentially followed by a further C-terminal deletion that would produce the LABD97 fragment (97 kDa) (Figure 8) [92,93]. Consequently, the inhibition of plasmin activation by gliptins may lead to the inappropriate plasmin-independent cleavage of BP180, resulting in the generation of neoepitopes along different domains [18].

#### 4.2.3. DPP4i-BP Autoantibodies and Epitope Spreading

Izumi et al. first identified a subtype of BP patients characterized by the presence of anti-BP180 full-length (BP180fl) autoantibodies without anti-BP180 NC16A autoantibodies. This subtype exhibited a non-inflammatory phenotype with less pronounced eosinophilic infiltration. Additionally, these patients were more likely to have been treated with DPP4i before BP onset [18].

Subsequent studies further describe the immunological profile of gliptin-associated bullous pemphigoid. All DPP4i-BP patients included in most studies present positive anti-BP180fl, whereas the positivity rate of anti-BP180 NC16A is notably lower (29–58%) [94,95,96]. A significant percentage of patients (79–86%) also present IgG reactivity against the LAD-1 domain, situated in the mid-portion of the extracellular domain of BP180 [95,96]. In contrast, in a recent case series involving 18 patients, all DPP4i-BP sera showed reactivity against the extracellular domain of BP180, but with a preference for the LABD97 domain over full-length BP180 [92]. Although anti-BP230 autoantibodies are rarely detected in DPP4i-BP, one case has been reported in which the only positive autoantibody was targeted against BP230, while BP180 fl and NC16A were both negative [97]. Moreover, IgE antibodies against BP230 and not BP180 have also been identified in DPP4i-BP patients, while no IgA reactivity was detected against BP180 or BP230 [98]. Finally, a case of DPP4i-BP with autoantibodies against the extracellular domain of α6β4 integrin, along with anti-BP180 NC16A antibodies, has been reported as well [99].

Although the presence of anti-BP180 NC16A autoantibodies is less common at the clinical onset of DPP4i-BP, several cases have been reported in the literature in which they become positive during the clinical follow-up. Mai et al. reported three cases of non-inflammatory DPP4i-BP with negative anti-BP180 NC16A that later evolved into an inflammatory phenotype, along with the appearance of positive anti-BP180 NC16A antibodies. It is notable that all these three patients had continued with the DPP4i treatment despite the BP diagnosis [100]. Takama et al. described the case of a patient who, even after discontinuing gliptin treatment, developed a positive response to anti-BP180 NC16A autoantibodies months after the diagnosis, coinciding with a clinical relapse [101]. In contrast, García-Díez et al. reported the case of a patient with an inflammatory subtype of DPP4i-BP and negative anti-BP180 NC16A autoantibodies who had discontinued gliptin treatment and was in clinical remission when anti-BP180 NC16A antibodies became positive [96]. These cases support the theory of epitope spreading in the pathogenesis of DPP4i-BP. It has been proposed that the discontinuation of DPP4i may lead to the restored cleavage of BP180 by plasmin within the NC16A domain, exposing the NC16A domain as an epitope for autoantibodies, as occurs in classic BP cases [101].

In classic BP, the initial autoantibody response commonly targets the NC16A domain on BP180. Subsequently, additional autoantibodies may emerge, typically directed towards other domains on BP180 or BP230. Conversely, in DPP4i-BP, the primary autoantibody response is directed against the extracellular domain of BP180, albeit in a distinct domain apart from NC16A, which likely involves the full-length molecule or the LAD-1/LABD97 domains. As a consequence of the phenomenon of epitope spreading, diverse additional autoantibodies may develop, even targeting the NC16A domain, despite the discontinuation of the drug and while the patient is in complete clinical remission (Figure 9) [102,103].

#### 4.2.4. Currently Described Immune and Pathogenic Mechanisms

While significant advancements have been achieved, the exact pathogenesis underlying the association of DPP4i and BP remains uncertain. However, recent research has demonstrated that DPP4i’s inhibition of plasmin reduces the degradation of the NC16A BP180 domain and may trigger the breakdown of immune tolerance [103].

As previously stated, Muramatsu et al. described that total Treg cells and all Treg subsets are increased in classic BP, while, in DPP4i-BP, neither the total Treg cell count nor their subsets seem to be increased. These results suggest that effector Treg cells with suppressive functions may be expanded in response to the inflammation environment seen in active classic BP, as effector Treg cells restrain autoreactive T cells. Then, the authors hypothesized that effector Treg cells in DPP4i-BP may not expand sufficiently in response to the autoreactive T cells due to the influence of DPP-4i intake, leading to the development of bullous lesions even in a mild inflammatory background [49].

It is well known that both IgG1 and IgG4 are the predominant autoantibodies in BP. In a case series of DPP4i-BP patients, all of them presented IgG1 autoantibodies against BP180, in contrast to IgG4-subclass autoantibodies, which were observed in only 38.9% of patients. This finding might be surprising since it is widely recognized that IgG1, and not IgG4, antibodies are able to activate the complement system and consequently initiate the classic inflammatory cascade. The predominance of IgG1 in DPP4i-BP suggests the involvement of the complement system in BP development in these patients, despite their typical association with non- or less inflammatory phenotypes [92].

Interleukin 6 (IL-6) is involved in both the pathogenesis and maintenance of BP [104]. Interestingly, Hung et al. recently reported that vildagliptin can stimulate the expression of IL-6 by keratinocytes in vitro, subsequently increasing its levels through a positive feedback loop. As a result, keratinocytes treated with DPP4i may supply sufficient IL-6 in the skin of DPP4i-BP patients, even if eosinophils and other inflammatory cells are reduced [105].

A recent study has reported that DPP4i drugs such as saxagliptin and sitagliptin promote the migration and epithelial–mesenchymal transition (EMT) of keratinocytes in vitro [106]. In parallel, BP180 is known to be involved in keratinocyte migration [107], so it has been hypothesized that DPP4i might affect keratinocytes in an EMT-dependent manner [108].

In vitro experiments by Nozawa et al. have demonstrated that DPP4 inhibitors upregulate both MMP9 and angiotensin-converting enzyme 2 (ACE2) via the angiotensin 1–7/Mas receptor (MasR) axis, and they postulate that this pathway may play a pivotal role in the development of BP. Lisinopril and MasR inhibitors effectively suppress the DPP4i-induced upregulation of MMP9, suggesting that the modulation of the renin–angiotensin system could stand as a therapeutic approach for DPP4i-BP. This intriguing in vitro finding is sustained by research databases, which indicate that the concomitant use of lisinopril in patients taking DPP4i can significantly reduce the incidence of DPP4i-BP [109].

Despite the progress made in understanding the underlying pathogenic mechanisms of DPP4i-BP, a crucial question remains unresolved: whether mere DPP4i exposure alone can trigger BP or if additional contributing factors are required [103].

### 4.3. Clinical and Immunological Distinct Features in Classic and DPP4i-Associated BP

Most studies indicate that DPP4i-BP may exhibit a certain male preponderance [71,74,78,79]; nevertheless, this trend is not consistently observed [73,74,110] and it must be noted that, in healthy individuals, gender does not have an impact on the pharmacokinetics of vildagliptin [111]. However, further research is needed to validate any gender-related disparities in BP susceptibility during DPP4i therapy.

Notable differences are mainly observed among studies conducted in Japan, which show distinct features like a non-inflammatory phenotype and atypical epitopes along BP180, as opposed to studies reporting typical features in European patients.

A non-inflammatory phenotype has been defined as those BP cases in which the Bullous Pemphigoid Disease Area Index (BPDAI) for urticaria/erythema activity is less than 10 points [86]. This non-inflammatory phenotype was the prominent DPP4i-BP presentation in Japanese and Chinese studies, with an estimated prevalence of 50–70% among all cases and a significantly lower urticaria/erythema activity BPDAI score [18,86,94,105]. This non-inflammatory predilection was also observed in one European case series [112]. However, this finding was not reproduced in any other study involving Caucasian patients, where the authors failed to find differences in urticaria/erythema activity BPDAI scores between DPP41-BP- and non-DPP4i-associated BP [88,98,110].

In the same vein, Asiatic studies report that DPP4i-BP skin biopsies show a milder eosinophilic dermal infiltrate accompanying the non-inflammatory phenotype [18,94,105,113], while European studies describe similar eosinophil counts in the upper dermis when compared to non-DPP4i-associated BP [88,114]. However, in the context of circulating eosinophils, two Israeli and Hungarian studies reflect the Japanese findings, as they also reported lower eosinophil counts in DPP4i-BP cases [112,115].

Patients with DPP4i-BP may present with a more severe bullous component, according to their significantly higher blister/erosion BPDAI score [105,110] or to the larger body surface area affected [115]. Other studies have also observed a trend towards a more severe disease, albeit lacking statistical significance [94,114]. Furthermore, some authors have reported a higher likelihood of trunk [110,115] and head involvement [115] in DPP4i-BP patients, but these findings have not been consistently observed in other studies. Other reports suggest that mucous membrane involvement may be more common in DPP4i-BP than in patients without prior DPP4i exposure [113,116]. Despite the isolated clinical differences reported in some studies, most European studies have concluded that there are no major clinical differences between DPP4i-BP and non-DPP4i-BP [117,118].

From an immunological perspective, Japanese studies report lower seropositivity of the anti-BP180 NC16A autoantibodies, with preferential reactivity against anti-BP180 full-length instead [86,94,95]. In contrast, European studies have reported similar positivity rates of anti-BP180 NC16A autoantibodies between DPP4i-BP and non-DPP41-BP [88,98,110,118]. However, while the detection rate of anti-BP180 NC16A autoantibodies might be similar, the average titers were significantly lower in the DPP4i-BP group when compared to non-DPP4i-BP cases [110,112].

#### 4.3.1. Effect of DPP4i Withdrawal

The available data regarding the effect of DPP4i withdrawal on the DPP4i-BP course present conflicting results.

A large multicentric, retrospective case–control study conducted by Benzaquen et al. clearly reported that DPP4i discontinuation led to partial or complete clinical remission in 95% of cases, with no further therapy needed to achieve and maintain this remission [74]. Other studies have also reported improved clinical outcomes, with most patients achieving remission after DPP4i withdrawal [72,116,119]. However, it is worth noting that most patients in these studies likewise received the standardized treatment protocol for BP, consisting of high-potency topical corticosteroids, with or without systemic corticosteroids. Since this standard treatment has consistently demonstrated high rates of complete remission in BP cases overall, it could mask the beneficial impact of gliptin withdrawal [8,118].

In contrast, other studies did not find significant differences in the prognosis or clinical response between patients who continued and discontinued DPP4i treatment [112,118]. Plaquevent et al. reported that the time required to achieve disease control (14–15 days) and the rate and timing of relapses were comparable in DPP4i-BP cases irrespective of gliptin discontinuation. Furthermore, they observed no differences in the incidence of relapses whether the DPP4i was stopped within the first month after BP diagnosis or at a later stage [118]. These findings do not support the previously suggested beneficial effect of gliptin withdrawal on the clinical outcomes of patients with BP.

It is currently unclear whether DPP4i-associated BP behaves as a true drug-induced BP, completely resolving upon drug discontinuation, or if it is indeed a drug-triggered or drug-aggravated BP, in which the drug acts as the immune response trigger, subsequently following an independent clinical course despite drug withdrawal. However, the long latency period between DPP4i initiation and BP development in most cases supports the drug-triggered hypothesis rather than the classical drug-induced cutaneous reaction [73].

As this issue continues to be a subject of debate, the most recent BP guidelines recommend, as a precautionary measure, at least considering gliptin withdrawal in patients with DPP4i-BP [8]. Given the potential severity of BP and the wide availability of alternative antidiabetic drugs, the current, safer approach is to replace gliptins with other diabetes medications. Combining gliptins with metformin has also been linked to an increased risk of BP, although such a risk has not been associated with metformin alone [81,116]. Therefore, as previously discussed in this review, in such cases, it is generally safe to continue metformin treatment, while careful consideration should be given to discontinuing the gliptin component [73].

#### 4.3.2. DPP4i-BP Clinical Subtypes

Based on the comprehensive scientific literature reviewed in this study, it is postulated that DPP4i-BP patients could be categorized into two distinct subtypes (Table 2).

**Drug-induced BP**. This subtype would represent the true drug-related BP and would appear de novo in patients with no prior genetic predisposition. Patients in this category exhibit distinctive features, including a non-inflammatory phenotype (Figure 10a), negative results for anti-BP180 NC16A autoantibodies and positivity for other epitopes within BP180, such as full-length autoantibodies, LAD-1 and LABD97. Additionally, they often display lower levels of tissue and peripheral eosinophilia. In these patients, discontinuation of DPP4i is considered mandatory to restrain the stimulus for the immune system and to ultimately achieve disease control.**Drug-triggered BP**. This subtype would occur in patients already predisposed to developing BP, and the initiation of DPP4i treatment would merely precipitate the onset of the bullous disease. These patients typically display the characteristic features observed in classic BP, including an inflammatory phenotype (Figure 10b) and the positivity and high titers of anti-BP180 NC16A autoantibodies, along with higher levels of peripheral and tissue eosinophilia. In this subgroup of patients, discontinuing DPP4i is also advisable to eliminate at least one of the contributing factors to BP etiopathogenesis. However, it is important to note that BP may persist even after discontinuing gliptin treatment.

## 5. Bullous Pemphigoid Associated with Antineoplastic Drugs

### 5.1. Immune Checkpoint Inhibitor-Associated Bullous Pemphigoid (ICI-BP)

Immunotherapy with checkpoint inhibitors targeting programmed cell death protein 1 (PD-1), programmed cell death ligand 1 (PD-L1) and cytotoxic T-lymphocyte-associated protein 4 (CTLA-4) has emerged as a highly effective treatment, leading to improved overall survival rates in a growing spectrum of malignancies [120]. Immune checkpoint proteins prevent the immune system from recognizing and eliminating cancer cells. Consequently, the use of immune checkpoint inhibitors (ICI) disrupts the tumoral evasion mechanisms, resulting in the increased activation of the immune system against the tumor [121].

However, this heightened immune activation is non-specific and can affect various organs, leading to so-called immune-related adverse events (irAEs). Cutaneous toxicity is one of the most prevalent irAEs [122], affecting roughly 30% of patients treated with anti-PD-1/PD-L1 and 50% with anti-CTLA-4 [121]. The combination therapy of PD-1/PD-L1+CTLA-4 demonstrates the highest incidence of irAEs, reaching up to 70% [120]. Although maculopapular eruptions are the most common type of cutaneous irAE, immunobullous eruptions are being increasingly reported in the literature. Among these, bullous pemphigoid is the most frequently observed phenotype, although cases of lichen planus pemhigoides, mucous membrane pemphigoid and pemphigus vulgaris have also been documented [121].

#### 5.1.1. Epidemiology of ICI-BP

The incidence of ICI-associated BP (ICI-BP) remains uncertain and varies across different studies, but it is estimated to occur in approximately 0.2 to 1% of patients undergoing treatment with PD-1/PD-L1 or CTLA-4 inhibitors [123,124,125]. Melanoma is the most frequently observed primary tumor associated with ICI-BP, followed by non-small-cell lung cancer (NSCLC) [121,123]. However, it must be noted that ICI have been approved and used for longer periods in melanoma patients compared to other tumor types, which could introduce potential confounding factors.

BP has been reported under treatment with both PD-1/PD-L1 and CTLA-4 drugs, including pembrolizumab, nivolumab, atezolizumab, durvalumab, cemiplimab, ipilimumab and combination therapies involving ipilumumab plus nivolumab or tremelimumab plus durvalumab. However, it is notably more prevalent in patients receiving anti-PD-1/PD-L1 agents (mainly pembrolizumab and nivolumab) than anti-CTLA-4 [121,123,126,127]. This discrepancy may be attributed to the distinct mode of action of each agent. PD-1 blockade is thought to operate during the effector phase of immune tolerance induction, aiming to restore the activity of quiescent T-regulatory cells. This reactivation in peripheral tissues may stimulate T cell cross-reactivity with self-antigens such as BP180 and BP230. In contrast, anti-CTLA-4 therapy is less likely associated with ICI-BP, possibly due to its preferential mechanism of action during the immune priming phase of tolerance induction, as well as its higher expression in lymphoid tissues compared to peripheral ones [128].

Unlike classic BP, which has a female predominance, ICI-BP is more frequent among male patients, accounting for approximately 71–77% of cases [121,124,129]. It has been reported that male patients with melanoma often display a higher tumor mutational burden and more immunogenic neoantigens, which can account for both the better survival outcomes and the increased susceptibility to irAEs including ICI-BP [130,131]. Similarly to general DABP, ICI-BP usually develops in younger patients compared to classic BP [121,129].

#### 5.1.2. Pathogenesis of ICI-BP

The exact pathogenesis of ICI-BP remains unclear [7]. While immune checkpoint inhibitors primarily target T cells, the central role in the pathogenesis of BP is played by B cells via their autoantibody production [132]. Therefore, it is widely believed that the autoimmune phenomena induced by PD-1/PD-L1 inhibitors involve the dysregulation of both B and T cells [133].

In addition to the BMZ, BP180 has also been identified on the surfaces of malignant melanocytic tumors and NSCLC. Initially, this finding raised suspicion about the potential paraneoplastic nature of BP, but accumulating evidence showing BP resolution after immunotherapy discontinuation and relapse upon rechallenge strongly supports a causal relationship between PD-1/PD-L1 inhibitors and BP [133].

##### T-Cell-Independent B Cell Activation

Not only T cells but also B cells express PD-1 and PD-L1, suggesting that treatment with anti-PD-1/PD-L1 agents may potentially activate pathogenic B cells independently of T cells (Figure 11) [7].

The “same-antigen theory” suggests that targeting BP180 on tumor cells can trigger cross-reactivity against the BP180 present in the BMZ [121,133]. The PD-1/PD-L1 signaling pathway inhibits the binding of tumor antigens, including BP180, to their receptors on B cells (BCR), suppressing B cell expansion. Consequently, PD-1/PD-L1 blockade amplifies the BCR response to BP180, resulting in the expansion of B cells and subsequent antibody production, ultimately leading to subepidermal cleavage in the dermoepidermal junction [133].

##### T-Cell-Dependent B Cell Activation

Another proposed hypothesis to explain ICI-BP is that the blockade of PD-1/PD-L1 may lead to the dysregulation of B-cell-regulatory T cells, consequently promoting antibody production (Figure 11). PD-1/PD-L1 signaling stimulates both T follicular helper (TFH) and T follicular regulatory (TFR) cells within follicular germinal centers. TFH cells are responsible for the selection and survival of B cells, which subsequently differentiate into either high-affinity antibody-producing plasma cells or memory B cells. In contrast, TFR cells inhibit TFH and B cells, thereby controlling undesired T-cell-mediated autoimmune responses. Inhibiting PD-1/PD-L1 negatively impacts both TFH and TFR subpopulations, resulting in the increased production of low-affinity plasma cells. These plasma cells can contribute to antibody-mediated autoimmune phenomena, including BP [133].

##### Autoantigens and Epitope Spreading

In patients with ICI-BP, anti-BP180 NC16A autoantibodies have been detected more frequently (70–80%) than anti-BP230 autoantibodies (7–29%) [121,129,133]. However, other autoantibodies targeting various epitopes have also been reported, including LAD-1 or C-terminal regions in BP180, desmoglein 1/3 and demoplakin 1/2, while up to 16% of patients exhibit no detectable autoantibodies [133].

Beyond cross-reactivity, the presence of autoantibodies against different epitopes observed in ICI-BP may be attributed to the epitope spreading phenomena, as seen in DPP4i-BP [101]. It has been suggested that autoantibodies may develop as a secondary response to a lichenoid reaction, which is one of the most common cutaneous irAEs. The interface dermatitis in lichenoid reactions may potentially expose antigens at the BMZ, rendering them susceptible to autoantibody development. This could also explain why BP stands out as the most prevalent ICI-induced bullous dermatosis, since hemidesmosomes are more exposed and prone to autoantibody formation following interface damage compared to desmogleins or intracellular molecules [121,133].

#### 5.1.3. Clinical Course and Management of ICI-BP

Unlike many cutaneous irAEs that occur shortly after ICI initiation, BP presents a longer latency period. According to a recent review by Merli et al., ICI-BP may develop after a median period of 26 weeks following the initiation of immunotherapy, ranging from 2 to a maximum of 209 weeks, and, in a small percentage, it can even develop after ICI treatment completion [121,123,129].

ICI-BP usually presents with a prolonged prodromal phase compared to classic BP, which is characterized by persistent pruritus and non-specific dermatitis [121]. Based on the existing literature, the median time between the initiation of ICI therapy and the onset of pruritus or a non-specific cutaneous eruption can be as short as 13–19 weeks. However, the development of bullae as in classic BP is frequently delayed, often occurring within the range of 28–39 weeks [134]. The fact that pruritus is one of the most common cutaneous irAEs makes the early diagnosis of ICI-BP during the pre-blistering phase quite challenging. To achieve this, a high awareness index is necessary, and any pruritic skin eruption that does not respond to topical corticosteroids should prompt consideration for a biopsy [127,135].

In terms of clinical presentation, ICI-BP typically displays a classic inflammatory phenotype, with tense blisters and erosions overlying erythematous plaques, commonly affecting more than 10% of the body surface area. The anatomical distribution of these lesions varies, but they most commonly appear on the trunk and extremities, with mucosal involvement in 17–20% of cases [121,129]. All these findings suggest that ICI-BP shares clinical features similar to classic BP. While most ICI-BP cases show histopathological findings similar to classic BP with eosinophil predominance, some neutrophil-predominant BP cases have also been reported in the literature [136]. No significant differences regarding DIF have been observed between classic and immunotherapy-associated BP [121].

Guidelines provided by oncologists for the management of irAEs recommend that the decision to stop immunotherapy should be based on the severity of BP. Given that the majority of ICI-BP cases can be classified as moderate to severe, a significant proportion of patients reported in the literature had to temporarily or permanently discontinue immunotherapy (58–75%) [123,125,127,129,133]. Notably, unlike traditional DABP, some cases of ICI-BP may develop or persist even after the cessation of immunotherapy, due to the prolonged immune activation associated with PD-1/PD-L1 inhibition or to the continual production of autoantibodies by activated B cells [125,129].

Lopez et al. suggest that immunotherapy could be reinitiated in patients whose BP can be effectively managed without systemic corticosteroids [137]. However, case series have reported relapse rates of approximately 50% following immunotherapy rechallenge, either with the same or alternative checkpoint inhibitors [123,133]. It is important to note that these case series involved only a limited number of patients who underwent rechallenge, making it difficult to draw definitive conclusions regarding the likelihood of recurrence after immunotherapy reinitiation.

Finally, while some retrospective studies have suggested a potential link between the onset of ICI-BP and an improved tumor response [138,139], the heterogeneity of the results and the small number of patients limit their potential applicability. Future prospective studies should be conducted to assess the tumor response in patients with ICI-BP to validate this hypothesis.

### 5.2. Other Antineoplastic Medications

Other antineoplastic medications have also been associated with the development of BP in single case reports.

Recombinant interleukin-2 (IL-2). Aldesleukin’s potential association with DABP is plausible due to the overexpression of IL-2 and its receptor in BP [140].Epidermal growth factor receptor (EGFR) inhibitors. Erlotinib has been associated with BP development in a patient with lung adenocarcinoma, possibly linked to the expression of EGFR in basal keratinocytes [141].Mammalian target of rapamycin (mTOR) inhibitors. Three cases of sirolimus- and everolimus-related BP in kidney transplant recipients have been reported in the literature. The pathogenic mechanisms could be attributed to the role of mTOR in the cell cycle or to the imbalance between cell-mediated and humoral immunity induced by mTOR inhibitors. Additionally, it might also be associated with factors related to the renal graft itself [142,143].BRAF inhibitors. Dabrafenib might induce a pemphigoid-like reaction with typical clinical and histological features, despite negative DIF [144].Cyclin-dependent kinase 4/6 (CDK4/6) inhibitors. BP has recently been linked with novel targeted antineoplastic therapies such as palbociclib, but the underlying pathogenic mechanisms remain unknown [145].

## 6. Bullous Pemphigoid Associated with Biologic Agents (BIBP)

Several biologic agents used to treat immune-related diseases have been identified as potential triggers of biologic-induced bullous pemphigoid (BIBP). These agents include TNF-α inhibitors like adalimumab, etanercept, efalizumab and infliximab; the anti-IL-12/IL-23 agent ustekinumab; and the anti-IL-17 and -23 inhibitors secukinumab and guselkumab [146,147]. As biologics are being more extensively used in the treatment of a wider range of immune-related disorders, there has been an increasing incidence of BIBP reported in recent years [147].

### 6.1. Pathogenesis of BIBP

#### 6.1.1. TNF-α Pathway

The pathogenesis of anti-TNF-α BIBP remains unclear, but three hypotheses have been suggested to explain the underlying mechanisms. Firstly, patients undergoing anti-TNF-α therapy may exhibit increased cell apoptosis, exposing novel autoantigens and subsequently triggering autoantibody formation [147]. In addition, anti-TNF-α agents may imbalance the cytotoxic T cell response, relieving the suppression of autoreactive B cells, resulting equally in the increased production of autoantibodies [148]. Lastly, TNF-α inhibitors may act as haptens, binding to and modifying the antigenic properties of BMZ components, which would be more susceptible to an immune attack [63].

Adding further complexity, anti-TNF-α agents have also been described as effective therapies for BP patients, in a somewhat paradoxical phenomenon [147]. This paradoxical phenomenon has been reported also in other immune-mediated dermatological conditions, such as pyoderma gangrenosum [149]. In BP, mast cells release TNF-α alongside various other mediators upon degranulation, which are responsible for recruiting neutrophils and eosinophils to the surrounding tissue [33]. Additionally, TΝF-α is found in higher concentrations in BP blister fluid compared to non-inflammatory blisters [150], and its circulating levels have been correlated with the severity and number of lesions in BP patients [151].

Liu et al. studied the impact of TNF-α on eosinophils and described that they can release both Th1 and Th2 chemokines depending on the surrounding microenvironment, particularly influenced by the presence of interferon (INF)-γ or IL-4 [152]. Consequently, it has been proposed that anti-TNF-α agents may have a dual role, as they can either treat or induce Th2-mediated diseases like BP according to the underlying immune profile [147].

#### 6.1.2. IL-17/23 Pathway

The IL-17/23 pathway is activated by several proinflammatory cytokines, including TNF-α. Therefore, IL-17, IL-12/IL-23 or IL-23 inhibitors may present a similar mechanism in the development of BIBP as with TNF-α blockers, which includes dysregulation in the Th1/Th2 immune response and the disinhibition of autoreactive B cells [147]. These findings suggest that the immunological state may shift from Th1 to Th2 dominance, leading to the release of Th2 chemokines like eotaxin, known in the pathogenesis of BP [153]. Notably, all reported cases of BP induced by IL-12/IL-23 or IL-23 inhibitors had a previous history of anti-TNF-α treatment [147] and the only anti-IL-17-related case had been previously treated with ustekinumab [154], thereby increasing the susceptibility to BP.

Conversely, the IL-17/IL-23 axis plays an essential role in the pathogenesis of BP, confirmed by the increased levels of both IL-17 and IL-23 in the sera and blister fluid of BP patients [45]. IL-17 induces neutrophils to release NE and MMP9, which can degrade BP180 and ultimately lead to dermal–epidermal cleavage [155]. This could explain the paradoxical phenomenon wherein there are cases of BP successfully treated by IL-17 and IL-12/IL-23 inhibitors [147].

### 6.2. Clinical Features of BIBP

The relationship between BP and biologic agents might be controversial due to the higher incidence of BP among psoriatic patients [156]. However, all reported cases of BIBP in patients with psoriasis were assessed as “probable” or “possible” according to the Naranjo scale and Karch–Lasagna algorithm [146].

Classic and biologic-induced BP usually present similar clinical features but, interestingly, BIBP itself may exhibit distinct features depending on the specific biologic agent involved. Husein-Elahmed et al. recently reviewed all BIBP cases in psoriatic patients and found that the mean latency time for BIBP to develop was shorter in individuals treated with anti-TNF-α agents (5 weeks) compared to those receiving ustekinumab (28 weeks). As a result, it is suggested that TNF-α inhibitors may cause a true drug-induced BP with rapid and widespread bullous eruption early in treatment, while IL-12/IL-23 blockers might cause a “drug-triggered BP”, characterized by a slower onset and sometimes a refractory course even after drug withdrawal and systemic treatment [146].

## 7. Bullous Pemphigoid Associated with Other Drugs

### 7.1. Bullous Pemphigoid Associated with Diuretics

Diuretics are commonly associated with BP development in the scientific literature. Multiple case reports and case–control studies have linked various diuretics with BP, including loop diuretics, thiazides and aldosterone antagonists. These three groups of diuretics contain sulfur groups within their molecular structures, thus suggesting a potential non-immunological mechanism for BP development involving interaction with the sulfhydryl groups present in the basement membrane zone [4]. An additional immunologic mechanism where these drugs act as haptens has also been suggested [157].

Loop diuretics (furosemide, bumetanide, torsemide). Some case–control and database studies suggest a link between furosemide and DABP development (OR 3.3–3.8) [72,158,159], while other studies do not find this association with loop diuretics [3,64]. Interestingly, furosemide-induced BP may primarily affects sun-exposed areas due to the drug’s well-known photosensitivity [160]. In one case report, switching from furosemide to bumetanide resulted in the complete clinical remission of BP [161], although bumetanide itself has also been associated with BP development in some patients [162,163]. Finally, torsemide has been linked to DABP in one case report due to its temporal relationship and structural similarity to furosemide [164].Aldosterone antagonists (spironolactone). Similar to furosemide, studies on the association between spironolactone and BP exhibit conflicting results [3,61,78,159]. However, a meta-analysis conducted by Liu et al., which included all previous case–control studies, reported a significant association between the use of aldosterone antagonists and BP, with a pooled OR of 1.75 (95% CI, 1.28–2.40; I2 = 4%) [64].Thiazides (hydrochlorothiazide). While some case reports suggest a clinical relationship between hydrochlorothiazide and DABP [157,165], larger case–control studies have not found a significant association [64].Acetazolamide. A single case report highlights a patient who experienced a relapse of well-controlled BP lesions one month after initiating acetazolamide. The authors suggest that this diuretic, commonly used in ophthalmology, may have triggered the BP’s recurrence due to its structural similarity to other diuretics like furosemide [166].

### 7.2. Bullous Pemphigoid Associated with Neurological Drugs

Multiple neurological drugs have been linked to BP development according to numerous clinical case reports, although only a limited number of case–control studies have been conducted. These neurological medications encompass neuroleptics (risperidone and flupenthixol), anti-depressants (escitalopram, fluoxetine and doxepin), antiepileptics (levetiracetam), dopaminergic drugs (amantadine) and anticholinergic agents (biperiden), as well as other medications, such as galantamine, gabapentin or teriflunomide [3,4,167,168]. In their meta-analysis, Liu et al. identified a significant association solely with anticholinergic and dopaminergic drugs [64].

Nevertheless, the main challenge when discussing neurological drug-related BP stems from the well-established association between neurologic and psychiatric disorders and the etiopathogenesis of BP itself. Although the studies referenced here employed multivariate analysis to mitigate their impact, the possibility of residual confounding factors persisting should be acknowledged [169]. Consequently, studies that further minimize the influence of concomitant neuropsychiatric disorders and that focus on understanding the responsible pathogenic mechanisms are needed before definitively establishing the risk between BP and the use of neurological drugs.

### 7.3. Bullous Pemphigoid Associated with Cardiovascular Drugs

Angiotensin-converting enzyme (ACE) inhibitors, angiotensin II receptor blockers (ARB) and calcium channel blockers (CCB) have been associated with the development or exacerbation of BP in several case reports, but these findings have not been consistently supported by larger case–control studies [4].

ACE inhibitors may induce BP through various mechanisms. Firstly, it has been proposed that ACE inhibition could activate the pro-inflammatory kinin system, potentially triggering BP. Secondly, ACE inhibitors might bind to lamina lucida proteins and modify their antigenic properties, acting as haptens [170]. Additionally, certain ACE inhibitors have shown acantholysis properties in vitro, suggesting that the subsequent exposure of BMZ antigens may contribute to BP development [171]. Specifically, captopril contains a thiol group within its molecular structure, so it can directly interact with sulfhydryl groups in the BMZ [172]. These proposed pathogenic mechanisms contrast with in vitro and population-based studies that show the decreased incidence of DPP4i-BP when lisinopril is used concomitantly with DPP4i [109].

DABP has also been reported in association with ARBs such as valsartan and losartan. It is hypothesized that they may induce BP in a similar way to ACE inhibitors through their acantholytic properties [173,174]. Moreover, since they present phenyl groups in their molecules, ARBs might be able to expose hidden epitopes following interaction with the BMZ [4].

Regarding CCB, DABP has been reported during treatment with dihydropyridines such as amlodipine and nifedipine [175,176]. Nifedipine has been demonstrated to induce acantholysis and subepidermal cleavage in skin models in vitro [177].

Finally, some case–control studies have suggested an inverse relationship between the use of lipid-lowering agents and the development of BP, pointing out that statins might have a protective role against BP due to their anti-inflammatory properties [3,178]. However, the prescription of statins is associated with cerebrovascular accidents, which are in turn involved in the etiopathogenesis of BP. Indeed, a larger recent study found no significant association between BP and statin use after adjusting for potential confounding factors [179].

### 7.4. Bullous Pemphigoid Associated with Antimicrobial Agents

Establishing a causal relationship between antimicrobial agents and BP is even more challenging compared to other drugs, since they are not chronically administered and obtaining a complete drug history can be especially difficult when latency periods are prolonged. There is only one case–control study that has reported the significantly higher use of antibiotics in general in BP patients compared to the control population [159], while other studies have failed to find such differences [3,61]. The remaining evidence regarding antibiotics-associated BP is largely based on single case reports, which may reflect casual rather than causal associations.

Several groups of antibiotics have been linked to BP, including penicillins, cephalosporins, quinolones, metronidazole, rifampicin and actinomycin, as well as antifungal agents such as terbinafine and griseofulvin. Penicillins and cephalosporins are sulfur-containing drugs, so they could induce BP through immune dysregulation affecting T-regulatory cells, or through a non-immunological mechanism by directly interacting with sulfhydryl groups in the dermoepidermal junction [4,7,67]. On the other hand, levofloxacin and ciprofloxacin are believed to act as haptens, binding to the proteins in the lamina lucida and modifying their antigenic properties [180,181].

## 8. Conclusions

Drug-associated bullous pemphigoid has been linked to nearly a hundred medications, with various mechanisms proposed for their potential role, including the two-step theory, molecular mimicry, hapten-like properties, immune dysregulation and direct non-immunologic actions. However, the majority of these associations are primarily based on temporal relationships, which do not necessarily establish causality. Confirming drug reactions through rechallenge is, in most cases, infeasible due to ethic concerns, except for immunotherapy agents, which can be often reintroduced and frequently result in the relapse of the skin lesions.

Larger and stronger evidence is available for gliptin-associated BP, as DPP4i are the most-frequently reported drugs associated with DABP. Consequently, their underlying pathogenic mechanisms have been extensively studied. Two DPP4i-BP phenotypes have been described: the true drug-induced one, with its own distinct features, and the drug-triggered one, resembling classic BP. Although not well defined, several pathogenic mechanisms have been suggested to explain DPP4i’s role in BP development, including impaired BP180 cleavage along with immune dysregulation mediated by distinct autoantibody profiles and the involvement of T-regulatory cells.

Immunotherapy-induced BP is gaining relevance due to the paradigm shift in antineoplastic treatment in oncology. While their exact pathogenic mechanisms remain unclear, it is widely believed that BP induced by PD-1/PD-L1 involves both B and T cell dysregulation. Biologic agents including anti-TNF-α and anti-IL17/23 have been described as both triggers and potential treatments for BP, in what appears to be a paradoxical phenomenon. Other medications associated with DABP, although with weaker evidence, include diuretics, antibiotics and neuropsychiatric and cardiovascular drugs (Table 3). In most cases, they are presumed to act as haptens, inducing an immunological response, or to directly disrupt the hemidesmosomal proteins along the dermoepidermal junction.

Future studies investigating genetic predisposition and molecular mechanisms should help us to better understand the clinical course of DABP, identify at-risk individuals and improve their prognosis and quality of life.

## 9. Future Directions

Future research on drug-associated bullous pemphigoid should depart from uncovering the pathogenic mechanisms that explain why such a diverse range of medications can trigger the same cutaneous disease. DABP presents a compelling etiopathogenesis, as its clinical course differs significantly from what is typically expected from drug reactions. Understanding why drug exposure can modulate the immune system, triggering a persistent immune reaction that remains even after discontinuing the culprit drug, presents a formidable challenge. Additionally, investigating HLA phenotypes and specific gene expression patterns among DABP patients could shed light on whether genetic predisposition plays a leading role in individual susceptibility to BP development following exposure to certain drugs. Finally, larger population-based observational studies are necessary to strengthen the link between DABP and other drugs beyond DPP4i and immunotherapy, as existing studies and reports lack robust evidence on this topic.

## Figures and Tables

**Figure 1 ijms-24-16786-f001:**
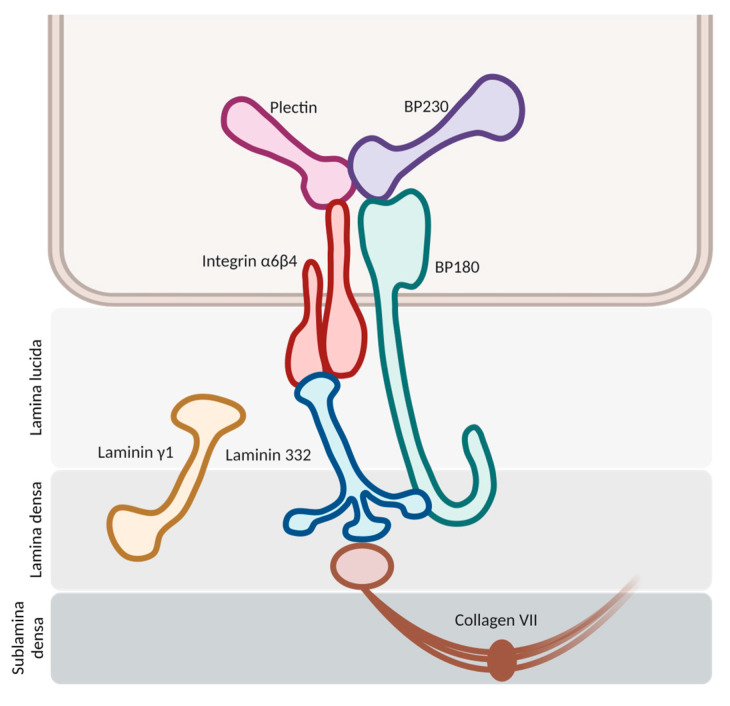
Schematic representation of hemidesmosomes in the basement membrane zone. The molecules of BP180 and BP230 stand as the main antigenic targets for autoantibody development in bullous pemphigoid.

**Figure 2 ijms-24-16786-f002:**
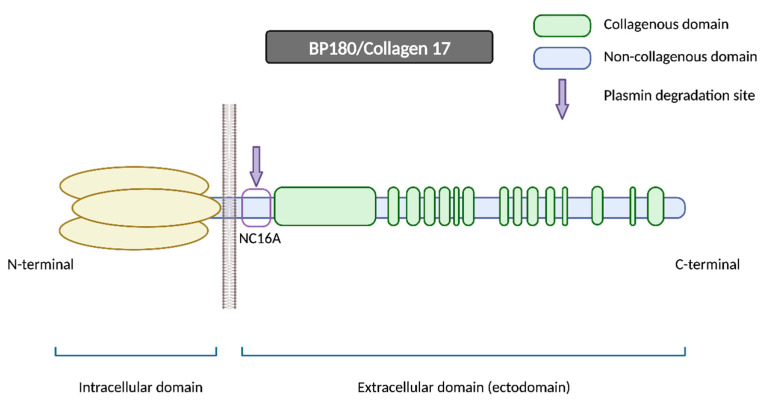
BP180/Collagen 17. BP180 is a transmembrane protein consisting of a globular intracellular domain at the amino-terminal and a large extracellular segment (or ectodomain) at the carboxyl-terminal. NC16A represents the juxtamembranous non-collagenous domain, and it contains the major pathogenic epitope for BP, in addition to being the primary site for plasmin and other serine proteases’ degradation. N-terminal: amino-terminal; C-terminal: carboxyl-terminal.

**Figure 3 ijms-24-16786-f003:**
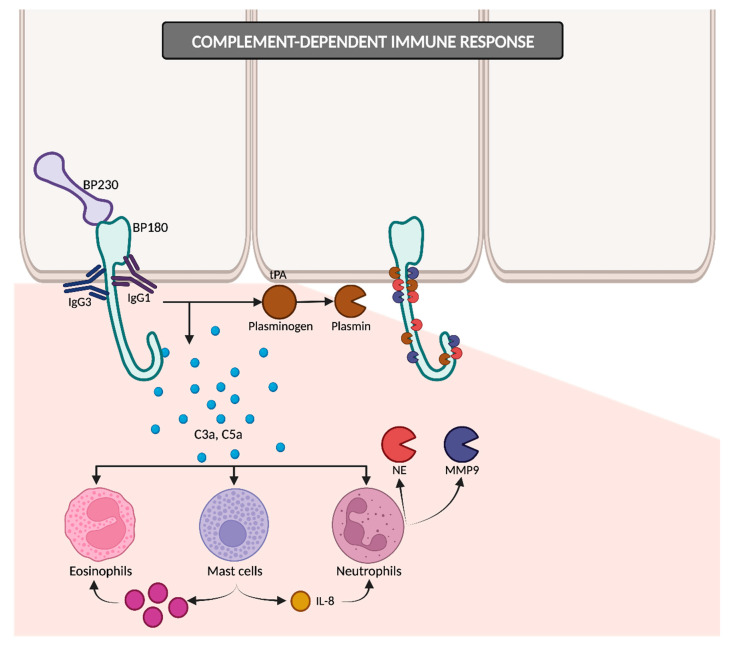
Complement-dependent immune response. IgG1 and IgG3 antibodies bind to BP180 and initiate the complement cascade, leading to eosinophils, mast cells and neutrophils’ recruitment and subsequent degranulation. IgG binding on keratinocytes may also trigger tPA release, converting plasminogen into plasmin. Plasmin, along with NE and MMP9, promotes dermal–epidermal cleavage. tPA: tissue-type plasminogen activator; NE: neutrophil elastase; MMP9: matrix metalloproteinase 9.

**Figure 4 ijms-24-16786-f004:**
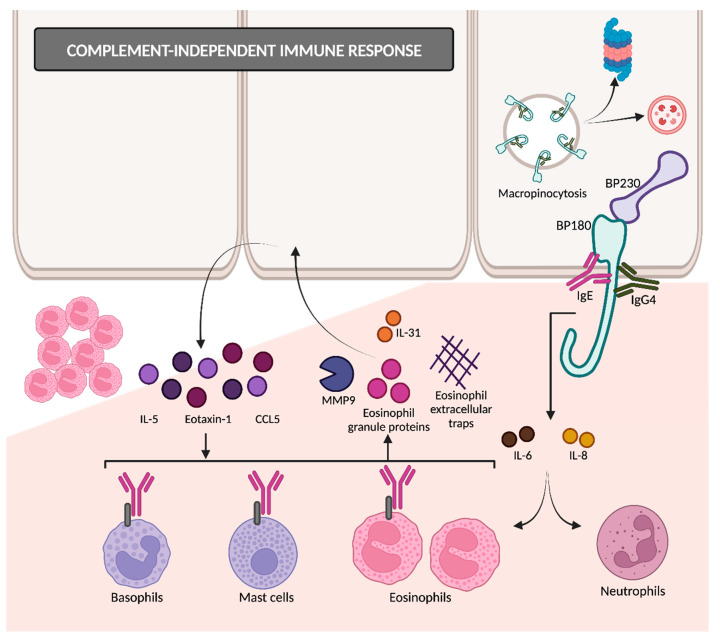
Complement-independent immune response. IgG4 and IgE are involved in BP pathogenesis through complement-independent mechanisms. Autoantibody binding to BP180 results in its internalization through the micropinocytosis pathway. The interaction between BP180 and autoantibodies triggers the release of several cytokines, ultimately attracting eosinophils and proteases that contribute to dermal–epidermal separation. MMP9: matrix metalloproteinase 9; CCL5: chemokine ligand 5.

**Figure 5 ijms-24-16786-f005:**
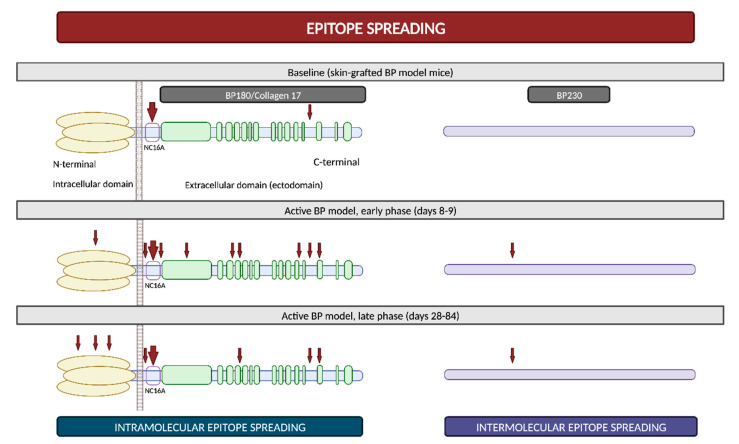
Epitope spreading. Epitope spreading in murine models according to the research conducted by Ujiie et al. Figure adapted from Ujiie et al. (2019) [53].

**Figure 6 ijms-24-16786-f006:**
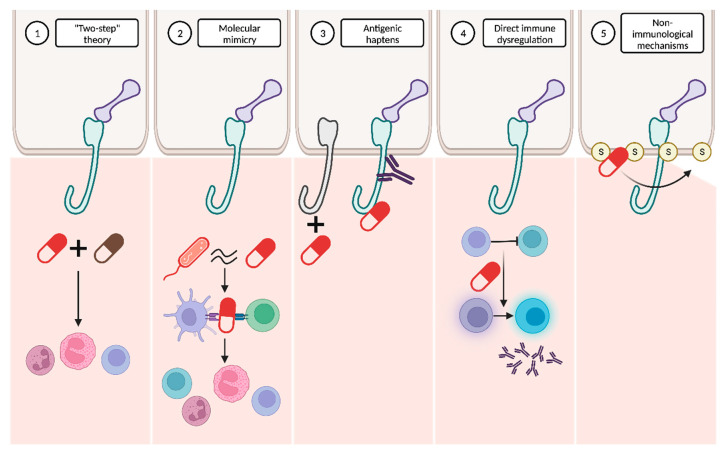
Proposed mechanisms to explain drug-associated bullous pemphigoid pathogenesis. (1) The “two-step” theory proposes that the interaction between two drugs may be necessary to initiate and amplify the immune response. (2) The molecular mimicry hypothesis suggests that the molecular similarity between certain drugs and microorganisms could trigger an immune response against the drugs. (3) Other drugs may act as antigenic haptens, modifying the antigenic properties of specific proteins. (4) Certain drugs may directly induce immune dysregulation. (5) Non-immunological mechanisms, involving interaction with sulfhydryl groups present in the dermoepidermal junction, are also plausible.

**Figure 7 ijms-24-16786-f007:**
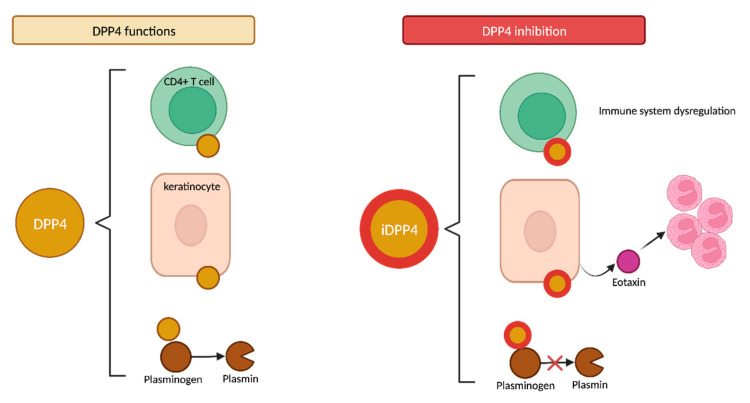
Schematic representation of DPP4 functions and, conversely, results following its inhibition. DPP4 is expressed in many tissues, such as CD4+ T cells and keratinocytes, and acts as a surface cell plasminogen that converts plasminogen into plasmin. As a result, DPP4 inhibition by gliptins may cause immune dysregulation, the release of eotaxin and other inflammatory mediators and the plasmin-independent cleavage of BP180.

**Figure 8 ijms-24-16786-f008:**
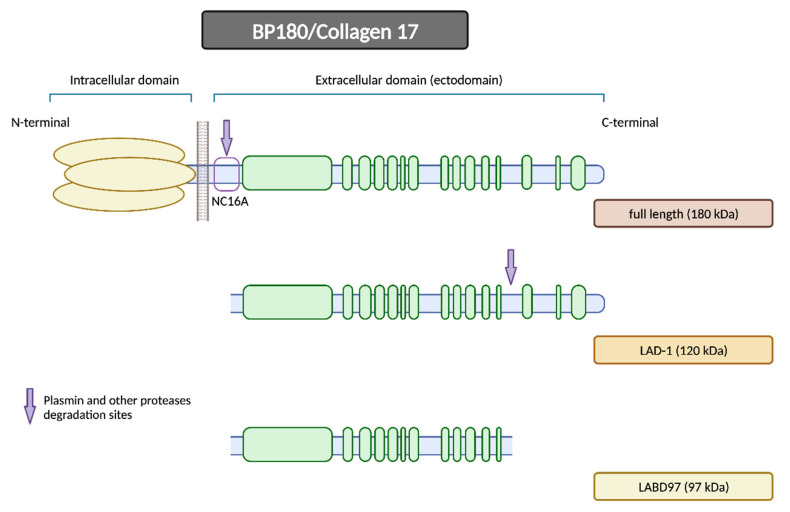
Plasmin and other proteases degrade the BP180 molecule within the juxtamembranous NC16A domain. The initial N-terminal cleavage results in the production of the LAD-1 fragment (120 kDa), which can be followed by a further C-terminal deletion that would result in the LABD97 fragment (97 kDa). Adapted from Mai et al. (2019) [92].

**Figure 9 ijms-24-16786-f009:**
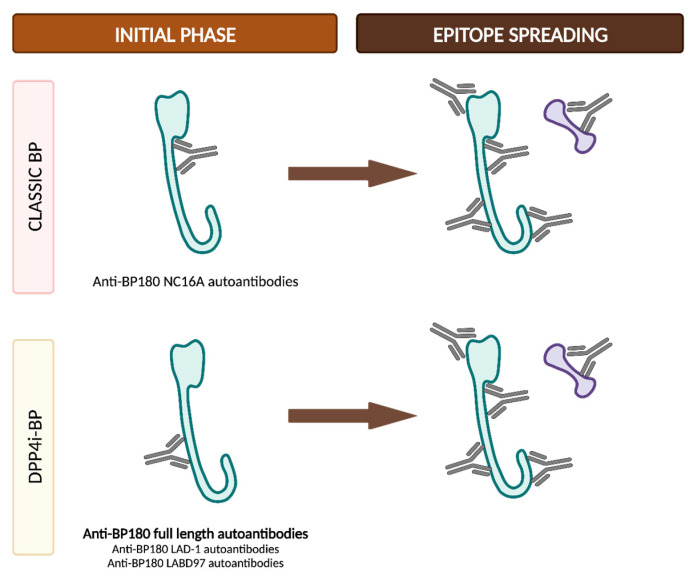
Presumed epitope spreading mechanisms in classic and DPP4i-associated BP.

**Figure 10 ijms-24-16786-f010:**
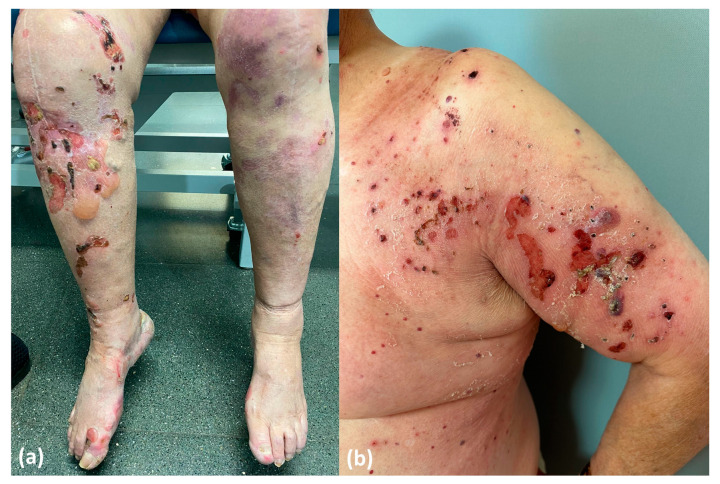
Clinical phenotypes in patients with DPP4i-BP: (**a**) Non-inflammatory phenotype, suggesting a drug-induced BP; (**b**) inflammatory phenotype, suggesting a drug-triggered BP. Commonly presumed epitope spreading mechanisms in classic and DPP4i-associated BP.

**Figure 11 ijms-24-16786-f011:**
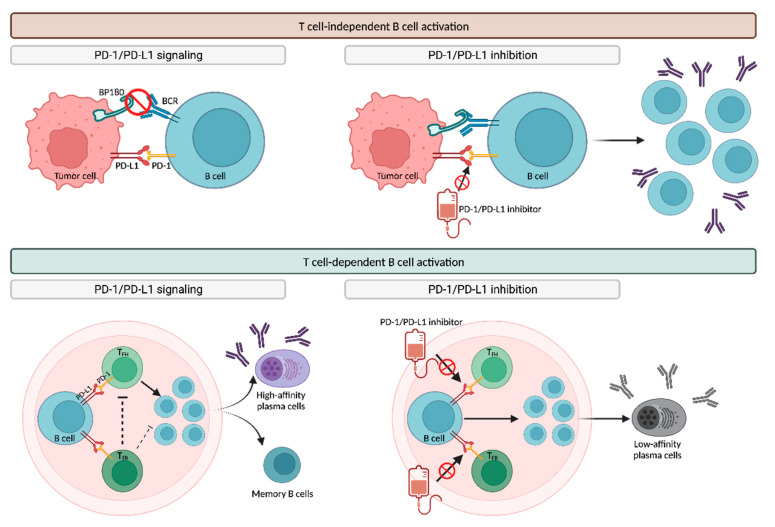
T-cell-independent and T-cell-dependent B cell activation in the pathogenesis of immune checkpoint inhibitor-associated bullous pemphigoid. Adapted from Tsiogka et al. (2021) [133].

**Table 1 ijms-24-16786-t001:** List of drugs that have been linked to bullous pemphigoid development according to the scientific literature. Adapted from Verheyden et al. [4].

Drugs Associated with Bullous Pemphigoid Development			
**Dipeptidyl peptidase 4 inhibitors**	**Diuretics**	**Immune checkpoint inhibitors**	**Biologic agents**
Alogliptin	Acetazolamide	Atezolizumab	Adalimumab
Anagliptin	Bumetanide	Cemiplimab	Efalizumab
Linagliptin	Furosemide	Durvalumab	Etanercept
Saxagliptin	Hydrochlorothiazide	Ipilimumab	Guselkumab
Sitagliptin	Spironolactone	Nivolumab	Infliximab
Teneligliptin	Torsemide	Pembrolizumab	Secukinumab
Vildagliptin			Ustekinumab
**Cardiovascular drugs**	**Neurological drugs**	**Antimicrobial agents**	**Anti-inflammatory drugs and salicylates**
Amiodarone	Amantadine	Actinomycin	Aspirin
Amlodipine	Doxepin	Amoxicilin	Azapropazone
Atenolol	Escitalopram	Ampicilin	Celecoxib
Captopril	Fluoxetin	Cephalexin	Ibuprofen
Enalapril	Flupenthixol	Ciprofloxacin	Mefenamic acid
Lisinopril	Gabapentin	Chloroquine	Mesalazine
Losartan	Galantamine	Dactinomycin	Metamizole
Nadolol	Levetiracetam	Griseofulvin	Phenacetin
Nifedipine	Risperidone	Levofloxacin	Sulfasalazine
Practolol	Teriflunomide	Metronidazole	Salicylazosulphapyridine
Rosuvastatin		Penicillin	
Valsartan		Rifampicin	
		Sulfonamide	
		Terbinafine	
**Other drugs**		**Topical drugs**	
Aldesleukin	Palbociclib	5-Fluorouracil	
Arsenic	Potassium iodide	Anthralin	
D-Penicillamine	Psoralens with ultraviolet A (PUVA)	Benzyl benzoate	
Dabrafenib		Coal tar	
Enoxaparin	Serratiopeptidase	Diclofenac	
Erlotinib	Sirolimus	Dorzolamide	
Everolimus	Tiobutarit	Iodophor adhesive band	
Omeprazole		Timolol	

**Table 2 ijms-24-16786-t002:** Suggested clinical and immunological subtypes of DPP4i-associated bullous pemphigoid.

	Drug-Induced BP	Drug-Triggered BP
	Different clinical entity	Similar to classic BP
Clinical phenotype	Non-inflammatory BP	Inflammatory BP
Latency period after DPP4i initiation	Shorter; higher risk in the first 3 months	Longer; even more than 6 years after initiation
Autoantibody profile	Anti-BP180 full length, associated or not with anti-BP180 LAD-1 and LABD97	Anti-BP180 NC16A
Eosinophils in skin biopsies	Decreased	Moderate infiltrate
Eosinophils in serum	Decreased	Augmented
DPP4i withdrawal	Mandatory	Recommendable
Clinical course	Less likely to persist after drug withdrawal	More likely to persist after drug withdrawal

**Table 3 ijms-24-16786-t003:** Summary of the drugs that are more and less likely associated with BP development according to the current scientific evidence.

Summary of Drugs Associated with Bullous Pemphigoid Development	
Strongly associated	Anecdotally associated
Dipeptidyl peptidase 4 inhibitors Immune checkpoint inhibitors Biologic agents (anti-TNF-α, anti-IL17/23)	Diuretics Neurological drugs Cardiovascular drugs Antimicrobial agents Anti-inflammatory drugs and salicylates Other drugs (D-Penicillamine, antineoplastic agents, topical drugs)

## Data Availability

The data presented in this study are available on request from the corresponding author.

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
