# Peer review of "From Molecular Insights to Clinical Perspectives in Drug-Associated Bullous Pemphigoid"

_ijms, 2023, doi:10.3390/ijms242316786_

Round 1

Reviewer 1 Report

Comments and Suggestions for Authors

My sincere compliments to the authors for this impressive review on the intriguing and interesting topic of drug-induced/triggered bullous pemphigoid.

The works is quite long but really educative, informative and readable, and provides an optimum overview on the topic, from pathophysiology to clinical features and management of these cases.

Scientific content is equilibrate and rigorous.

Table are figures are nice, complete and informative.

English and quality of writing is perfect.

I can consider some really minor revisions:

1) Figure 6 could be amelioreted with a short legend describing in brief the depicted mechanisms.

2) The authors could gently cite this work on the TNF-alfa paradoxical reaction (doi: 10.3389/fmed.2023.1197273)

It could be placed in the sub-chapter 6.1.1, lines 907-908, adding or specifying the increased expression of interferon gamma by plasmocytoid dendritic cells as an ancillary effect (along with other mechanism reported).

3) A riepilogative table could be inserted summarizing the strongest associated drugs and the anedoctical ones (as correctly described in the conclusion).

Compliment again for the great work.

Author Response

1) Figure 6 could be amelioreted with a short legend describing in brief the depicted mechanisms.

Response 1: that is a great suggestion. We have included a short legend in the figure caption to make it more understandable (highlighted in red, page 11, lines 364-371).

2) The authors could gently cite this work on the TNF-alfa paradoxical reaction (doi: 10.3389/fmed.2023.1197273)

It could be placed in the sub-chapter 6.1.1, lines 907-908, adding or specifying the increased expression of interferon gamma by plasmocytoid dendritic cells as an ancillary effect (along with other mechanism reported).

Response 2: thank you for the recommended bibliography. We have included the work as another example of paradoxical phenomenon (highlighted in red, page 24, lines 921-923).

3) A riepilogative table could be inserted summarizing the strongest associated drugs and the anedoctical ones (as correctly described in the conclusion).

Response 3: Thank you very much for the suggestion. We have included a table in the conclusion summarizing the drugs that show the most and least scientific evidence and we believe that this helps to make it much clearer for the readers. (highlighted in red, page 27, lines 1088-1090).

Reviewer 2 Report

Comments and Suggestions for Authors

This is a well-prepared and extremely comprehensive review article for drug-associated bullous pemphigoid (DABP).  This article summarized all types of DABD and deeply analyzed various aspects, including clinical and histopathological findings, epidemiology and pathogenesis in this condition.  Thus, this article provides all information for DABD and should be important for both clinical and investigative aspects.  The manuscript is well written, figures and tables are well prepared, and English is in general good.  However, I have several minor comments, which is described below.

(1) In the abstract section, the authors should use the term "DPP4 inhibitor", because this term is used more frequently than the term "gliptin" in the text.

(2) The positions of some molecules in the figure 1 are somehow incorrect.  They should be carefully corrected.   In addition, the domains of collagen VII should also be explained in the figure legend.

(3) The N- and C-terminals should be explained in the figure legend of the figure 2.

(4) The phrase "The closest NC domain to the amino-terminal, named NC16A" is somehow confusing and may be improved.

(5) The titles may be added at the first places of the figure legends for the figures 2-5.

(6) The sentences in the figure legend and reference number for the figure 5 may be moved inside the text 

Comments on the Quality of English Language

(1) The phrase "from Molecular to Clinical Aspects" is somehow incorrect and may be better to be re-written.

(2) Abbreviations are used incorrectly at some places, which should be carefully corrected.  For example, full spell for ES is not shown at line 267.  "BP" and "bullous pemphigoid" are incorrectly used at page 12.

Author Response

(1) In the abstract section, the authors should use the term "DPP4 inhibitor", because this term is used more frequently than the term "gliptin" in the text.

Response 1: thank you for the suggestion. We have already corrected it in the abstract (highlighted in red, page 1, lines 13 and 16).

(2) The positions of some molecules in the figure 1 are somehow incorrect.  They should be carefully corrected.   In addition, the domains of collagen VII should also be explained in the figure legend.

Response 2: Thank your for pointing out that the placement of some of the hemidesmosomal molecules were not exactly correct. We have already corrected their position and included the new “Figure 1” in the new manuscript. However, we have not explained the structure of collagen VII because we believe that if we explained the domains of collagen VII, we would also have to explain the molecular structure of the other proteins that are part of the hemidesmosomes, and maybe this could distract the reader from what is really important for bullous pemphigoid, which are BP180 and BP230. In any case, if you still consider that explaining it is important for readers, let us know and we will do it.

(3) The N- and C-terminals should be explained in the figure legend of the figure 2.

Response 3: we have included the explanation of these abbreviations in the figure legend (highlighted in red, page 3, line 98).

(4) The phrase "The closest NC domain to the amino-terminal, named NC16A" is somehow confusing and may be improved.

Response 4: we have re-written that sentence to make it more clear for readers: “NC16A domain consists of an extracellular juxtamembranous region and contains the major pathogenic epitope for BP” (highlighted in red, page 3, lines 99-100).

(5) The titles may be added at the first places of the figure legends for the figures 2-5.

Response 5: thank you for the suggestion. We have included the titles of the figures on figure legends 2-5 (highlighted in red).

(6) The sentences in the figure legend and reference number for the figure 5 may be moved inside the text.

Response 6: we have deleted the long explanation of the figure legend number 5. However, we have maintained the reference number in the figure legend because our figure is just an adaptation from the experiment conducted by Ujiie et al., and we would like readers to easily find it when reading our review.

Comments on the Quality of English Language

(1) The phrase "from Molecular to Clinical Aspects" is somehow incorrect and may be better to be re-written.

Response 1: we have re-written the title of our review. The new title that we propose is “From Molecular Insights to Clinical Perspectives in Drug-Associated Bullous Pemphigoid”. We hope it sounds better both for reviewers and readers.

(2) Abbreviations are used incorrectly at some places, which should be carefully corrected.  For example, full spell for ES is not shown at line 267.  "BP" and "bullous pemphigoid" are incorrectly used at page 12.

Response 2:

ES was not fully spelled at line 267 because it was previously spelled some lines before in line 248 (according to the new manuscript). However, we have re-written it as the whole words rather than the abbreviation to avoid reader’s misunderstanding (highlighted in red, page 7, line 264).

On the other hand, I am afraid we are not able to find where “BP” and “bullous pemphigoid” are incorrectly used at page 12. If you could please provide us with the number of the line where it is incorrect, we will be very glad to correct it.
